# Systematic review and tools appraisal of prognostic factors of return to work in workers on sick leave due to musculoskeletal and common mental disorders

Patrizia Villotti[1,2][☺]*, Ann-Christin Kordsmeyer[3], Jean-Sébastien Roy[4,5],
Marc Corbière[1,2], Alessia Negrini[6], Christian Larivière[6☺]

1 Department of Education and Pedagogy–Career Counseling, Université du Québec à Montréal, Montréal, Canada, 2 Centre de recherche de l'Institut universitaire en santé mentale de Montréal, Montréal, Canada, 3 Institute for Occupational and Maritime Medicine (ZfAM), University Medical Center Hamburg-Eppendorf (UKE), Hamburg, Germany, 4 School of Rehabilitation Sciences, Faculty of Medicine, Université Laval, Quebec City, Canada, 5 Centre for Interdisciplinary Research in Rehabilitation and Social Integration, Quebec, Rehabilitation Institute, Quebec City, Canada, 6 Institut de recherche Robert-Sauvé en santé et en sécurité du travail, Montréal, Canada

☺ These authors contributed equally to this work.
* villotti.patrizia@uqam.ca

**Data Availability Statement:** All relevant data are within the manuscript and its Supporting Information files.

## Abstract

With the overall objective of providing implication for clinical and research practices regarding the identification and measurement of modifiable predicting factors for return to work (RTW) in people with musculoskeletal disorders (MSDs) and common mental disorders (CMDs), this study 1) systematically examined and synthetized the research evidence available in the literature on the topic, and 2) critically evaluated the tools used to measure each identified factor. A systematic search of prognostic studies was conducted, considering four groups of keywords: 1) population (i.e., MSDs or CMDs), 2) study design (prospective), 3) modifiable factors, 4) outcomes of interest (i.e., RTW). Studies showing high risk of bias were eliminated. Tools used to measure prognostic factors were assessed using psychometric and usability criteria. From the 78 studies that met inclusion criteria, 19 (for MSDs) and 5 (for CMDs) factors reaching moderate or strong evidence were extracted. These factors included work accommodations, RTW expectations, job demands (physical), job demands (psychological), job strain, work ability, RTW self-efficacy, expectations of recovery, locus of control, referred pain (back pain), activities as assessed with disability questionnaires, pain catastrophizing, coping strategies, fears, illness behaviours, mental vitality, a positive health change, sleep quality, and participation. Measurement tools ranged from single-item tools to multi-item standardized questionnaires or subscales. The former generally showed low psychometric properties but excellent usability, whereas the later showed good to excellent psychometric properties and variable usability. The rigorous approach to the selection of eligible studies allowed the identification of a relatively small set of prognostic factors, but with a higher level of certainty. For each factor, the present tool assessment allows an informed choice to balance psychometric and usability criteria.

**Funding:** This research project was Co-funded (grant #2020-0028 to PV, CL, AN, JSR, and MC) by the Institut de recherche Robert-Sauvé en santé et en sécurité du travail (IRSST: https://www.irsst.qc.ca/en/) of Quebec (Canada) and the Réseau provincial de recherche en adaptation-réadaptation (REPAR: https://repar.ca/). These funders were not involved in any step of the research or publication process.

**Competing interests:** The authors have declared that no competing interests exist.

## Introduction

Contemporary societies view employment as the standard and essential aspect of adulthood and full citizenship. Consequently, work is central to a person's identity, social role, community status and overall wellbeing [1]. People with disabilities or decreased functional abilities, whether temporary or permanent, may face challenges in entering, staying in, or re-entering the workforce [2, 3]. In particular, musculoskeletal disorders (MSDs, such as low back and neck pain, joint pain, tendonitis, carpal tunnel syndrome) and common mental disorders (CMDs, such as depression, anxiety, mood disorders) are leading contributors to work disability and to the global need for rehabilitation services worldwide [4, 5]. This entails important economic implications, as well as hefty social and personal burdens [6, 7]. Return to work (RTW) is often used as an indicator of recovery, and it is consistently reported to be associated with improved mental and physical health, increased quality of life, and enhanced social functioning [8]. To identify the factors that contribute to RTW after the onset of a physical or mental disability has thus become a priority in all industrialized countries. Of particular interest are those factors promoting or hindering RTW that are *modifiable* throughout a targeted intervention. Modifiable factors can be modified or adjusted through RTW, disability prevention programs, or other clinical and occupational interventions that aim to increase or maintain individual health and capabilities [9–11].

According to the disability prevention management model [12], predictive factors of RTW can be divided, in theory, according to four systems, namely the personal, workplace, healthcare and compensation systems. Unfortunately, factors related to the healthcare and compensation systems have scarcely been studied using only longitudinal observational studies. Factors that have been more extensively studied can be divided into organizational factors, such as work accommodations, and personal factors that can be work-related, such as RTW expectations, or non-work-related, such as recovery expectations. These factors, also called contextual factors, have been identified and classified in categories that are relevant to the occupational rehabilitation domain, extending the well-known International Classification of Functioning (ICF) framework [13], as illustrated in Fig 1.

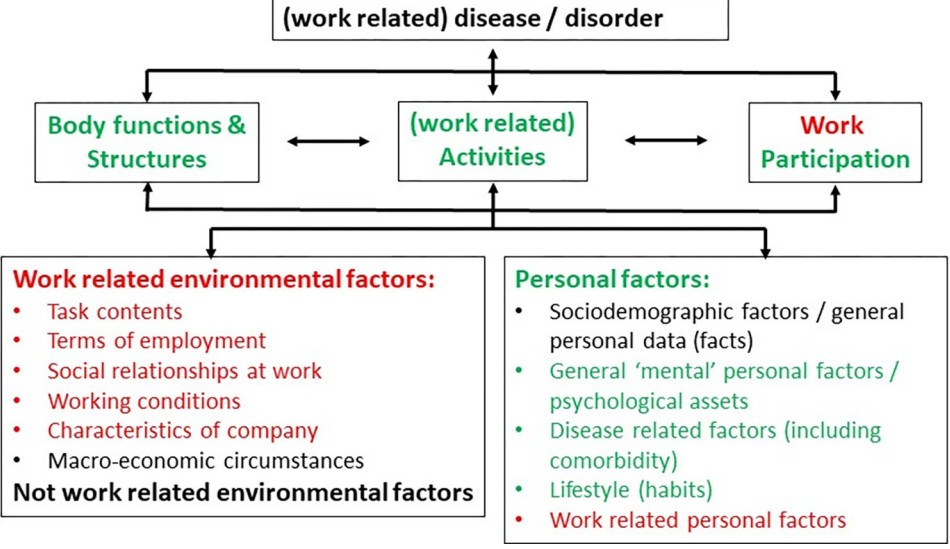

**Fig 1. International Classification of Functioning framework, adapted from [13].** Prognostic factors from the classification categories identified in red characters are environmental (here organizational) and personal work-related modifiable factors, while personal non-work-related modifiable factors are identified in green characters.

Prognostic factors contributing to RTW have been the subject of several reviews [10, 11, 14–23]. However, there are great variability in the identified factors, which can become problematic when designing interventions. This variability may primarily depend on the health condition [14, 18] and whether RTW or sustainable RTW are considered as outcomes. However, much variability may also be explained by the design and quality of the included studies and more specifically, by the way to manage the different risks of bias (RoB). Another challenge for researchers and practitioners is how the prognostic factors are evaluated/measured, because different tools exist to assess a same concept [17, 20].

A common strategy to cumulate the scientific evidence on a given factor is to merge concepts that are closely related. However, creating factors from the merging of concepts that are a little too distinct may potentially lead to misleading conclusions. Therefore, the merging of concepts requires to look at the items of each tool to make sure they tap on the same broader concept, at least in terms of face validity. In addition, the quality of tools used to measure the different concepts or factors might be influential regarding their ability to predict RTW outcomes. In fact, the prognostic factor measurement is one of the domains considered when assessing the RoB in studies of prognostic factors [24]. To the authors' opinion, when the number of studies using different tools are numerous enough for a given concept, instead of only merging the evidence across all the tools available, it might also be relevant to verify whether the corresponding factor becomes predictive only for specific tools. This strategy might be relevant especially for non-work-related factors as they have been much more studied than organizational or personal work-related factors, considering that the disease treatment paradigm (or biomedical model) has been introduced much earlier than the disability prevention paradigm (or biopsychosocial model) [12]. Ideally, health care professionals and researchers should measure factors predictive of RTW using standardized tools. However, for each of these factors, the tools that have been used in studies showing a relationship with RTW outcomes are generally not explicitly presented. Assessing their psychometric qualities and usability would allow practitioners and researchers to make an informed choice of available tools based on their practice/research setting.

Two literature reviews and tool appraisals of factors predicting RTW in workers on sick leave due to MSDs and CMDs have been recently published, more precisely on organizational factors [17] and personal work-related factors [20]. Despite including only studies using a prognostic design in these reviews to establish a stronger predictive association with RTW, and conducting a systematic search across databases, there is a lack of quality assessment of the included prognostic studies. A more stringent approach by rejecting studies with a high RoB and then basing quality of evidence on the quantity of studies alone is thus needed. It is hypothesized that this would help reducing "noise" and therefore get a clearer picture of the prognostic factors of RTW.

With the overarching objective of providing implication for clinical and research practices regarding the identification and measurement of modifiable predicting factors for RTW in people with MSDs and CMDs, this study 1) systematically examined and synthetized the research evidence available in the literature on the topic, and 2) critically evaluated the tools used to measure each identified factor. A particular effort has been made to define distinct concepts related to predictive factors (e.g., coping can be separated into coping styles, coping cognitive strategies and coping behavioural strategies). Moreover, when possible, the evidence has been separated from specific tools measuring the same concepts (e.g., perceived disability can be measured with the Oswestry, Roland-Morris or SF-36 questionnaires).

## Materials and methods

### Study design

A systematic literature review was conducted to identify prognostic factors associated with RTW among workers on sick leave due to MSDs and CMDs. The identified prognostic factors were evaluated based on their level of evidence, categorized as strong, moderate, limited, inconsistent, insufficient, or non-significant. Subsequently, the tools used to assess each identified prognostic factor, which exhibited moderate to strong evidence, were systematically described and evaluated. This evaluation encompassed an assessment of the psychometric properties of the tools, including reliability and validity, as well as their usability criteria.

### Phase 1 –Identification of the prognostic factors of RTW

**Search strategy.** Under the supervision of a research librarian, three databases (i.e., PubMed, CIHAHL and PsycINFO) were searched from their inception to January 19th, 2023, to retrieve articles on organizational, personal work-related, as well as personal non-work-related prognostic factors of RTW. Four groups of keywords served to identify potential articles: 1) disability condition; 2) outcome of interest; 3) prognostic factors; 4) study design. The search strategy is reported in S1 File. Additional articles were extracted from bibliographic references of included articles or from reviews on this topic.

In this review, RTW as an outcome was conceptualized either as the 1) being or not returned to work at the follow-up, or 2) the time taken to RTW or the duration of work absence. Other RTW measures, collected from employer/insurance databases, such as 1) being or not on wage replacement benefit, or 2) time to benefit suspension/time to claim closure, were also retained as acceptable surrogates of RTW in the present review. Sustainable RTW was not retained as a RTW outcome as the corresponding prognostic factors might be different.

**Study selection.** Studies were included if they met the following criteria: 1) they had a prospective design; 2) participants were workers with MSD or CMD or, for mixed population studies, at least two thirds (67%) of the workers were suffering from MSDs and/or CMDs; 3) participants were fully or partly on sick leave at baseline; 4) RTW or duration of sickness absence was analysed as an outcome; 5) factors were measured as predictors of the RTW outcome in multivariate analyses controlling for age and sex; 6) were published in English or French. Articles displaying a high RoB were excluded. While it is common practice to assess the quality of evidence for a particular factor by considering the quantity (number, proportions) and the quality of included studies, we propose a more rigorous approach by excluding studies with a high RoB and subsequently basing the assessment of evidence quality on the quantity of studies alone.

Study selection was conducted by using COVIDENCE software. After removing duplicates, two trained evaluators (MSc or PhD students) independently screened each title and abstract. Two additional authors (PV and CL) double checked 30% of the references. All relevant full text articles were then obtained and screened by three independent authors (PV, A-C K, CL) to determine if they met the inclusion and exclusion criteria. Discussion among the three authors was undertaken to reach full agreement when the inclusion of an article was uncertain.

**Methodological quality assessment of prognostic studies.** Risks of bias were evaluated with the Quality In Prognosis Studies (QUIPS) tool [24]. As outlined in Table 1, the QUIPS comprises six bias domains: 1) study participation, 2) study attrition, 3) prognostic factor measurement, 4) outcome measurement, 5) study confounding and 6) statistical analysis and reporting. To our knowledge, there is no guideline to determine the overall RoB of a given

**Table 1. Six bias domains of the Quality In Prognosis Studies (QUIPS) tool to assess study quality, four of them (1, 4, 5 and 6) being used to exclude studies.**

| Bias domains | Exclusion criteria |
|---|---|
| 1. <u>Study participation (6 items)</u>: The study sample represents the population of interest with regard to key characteristics, sufficient to limit potential bias to the results. | Workers on partial or full sick leave due to a MSD or CMD represent less than 2/3 of the total sample. |
| 2. <u>Study attrition (5 items)</u>: Loss to follow-up is unrelated to key characteristics (that is, the study data adequately represent the sample), sufficient to limit potential bias. | N.A.* |
| 3. <u>Prognostic factor measurement (6 items)</u>: The prognostic factor of interest is adequately measured in study participants, sufficient to limit potential bias. | N.A.*<br>Note: This review will be followed by the appraisal of tools of factors that will show strong or moderate quality evidence. Consequently, tools with questionable psychometric characteristics (except for predictive validity) were not eliminated at this step. This is to offer different options (tools) to users, showing different psychometric and usability characteristics. |
| 4. <u>Outcome measurement (3 items)</u>: The outcome of interest is adequately measured in study participants, sufficient to limit potential bias. | Note: Partial of full return to work (binary variable) or time to return to work (continuous variable) are easy to assess as they are conceptually simple. Labelling such as "increased working time" is equivalent. |
| 5. <u>Study confounding (7 items)</u>: Important potential confounders are appropriately accounted for, limiting potential bias with respect to the prognostic factor of interest. | Age, sex and treatment (in RCTs) as confounding variables were not considered at least in a preliminary step, showing their lack of relationship with the outcome, or not considered in the final multivariate model. |
| 6. <u>Statistical analysis and reporting (1 item)</u>: The statistical analysis is appropriate for the design of the study, limiting potential for the presentation of invalid results | There were no statistical analyses allowing to adjust for age and sex (and treatment in RCTs) in preliminary (to reduce one by one the number of potential prognostic factors) or final multivariate analyses. |

*N.A.: Not applicable as no study was excluded based on this risk-of-bias domain

CMD: common mental disorder; MSD: musculoskeletal disorder; RCT: randomized controlled trial.

study, and the developers of QUIPS recommend against the use of a summated score for overall study quality. Other research groups have suggested different thresholds [19], based on the total score, which basically reflects the number of domains met. We thus determined that a study would have a low RoB when all 6 bias domains were rated as having low RoB, a moderate RoB when 5 out of 6 bias domains were rated as having low RoB, and a high RoB when less than five bias domains were rated as having low RoB. As mentioned, studies with a high RoB were excluded from this review, more specifically the ones showing a high RoB for at least one of four specific bias domains identified in Table 1.

**Methodological quality assessment of evidence on risk factors.** For the assessment of the quality of evidence regarding risk factors, we adapted an approach similar to GRADE [25], a widely recognized method for evaluating the quality of evidence pertaining to interventions. In GRADE, evidence derived from studies with the highest level of evidence (e.g., RCT for intervention studies) is initially classified as high-quality evidence. Nevertheless, confidence in the evidence may be downgraded from high to moderate, and subsequently to low and very low, for different reasons (e.g., inconsistency in relative effects). In the case of assessing the quality of evidence concerning risk factors, it was determined that prognostic studies using a prospective design represented the highest level of evidence.

**Data extraction.** Trained evaluators (MSc or PhD students) first extracted the data for characteristics of the participant sample, type of outcome, time of outcome follow-ups, name

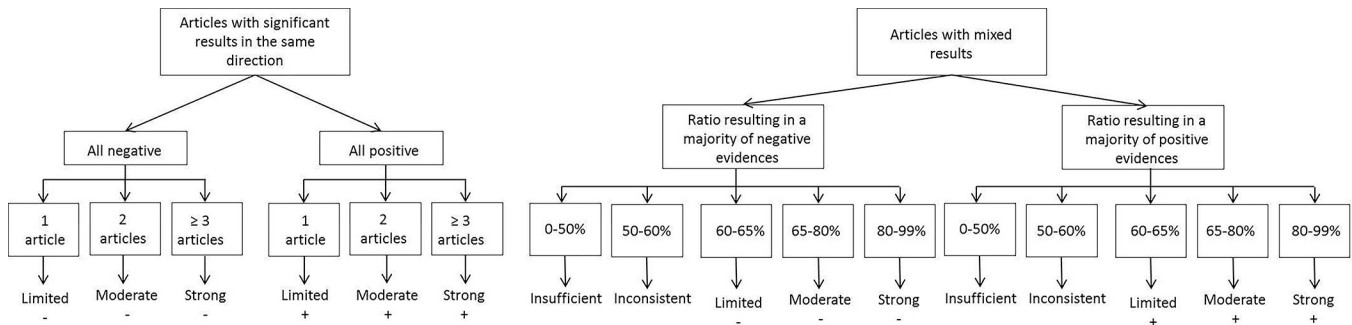

**Fig 2. Rules applied to synthesize the evidence (adapted from Gragnano et al. [17]).**

of the factor, whether it predicts RTW or not, and the tools used to measure these factors. Because of the complexity of the procedure adopted to identify the different concepts/factors and the understanding of multivariate statistical analyses, the entire database was revised by three researchers (PV, A-C K, CL). The present review focused solely factors measured at baseline.

**Data analysis.**  To adopt a common language, factors were first labelled based on the work of Heerkens and colleagues [13], whom have extended the ICF terminology to better reflect contextual factors, more precisely work-related environmental and personal factors. Partly, we elaborated the terminology further to include factors that were not labelled in the paper of Heerkens and colleagues [13] (e.g., differentiation of social support from various sources like supervisors, coworkers, families or from all sources).

In some studies, outcomes were measured at multiple follow-up points, or different tools were employed to measure the same concept. In such instances, the tools with significant findings were initially selected. If multiple tools led to significant findings, the most psychometrically sound tool was selected. In cases with multiple follow-up times, a factor was considered predictive if it demonstrated statistical significance in at least one of the follow-up assessments.

The factors were classified as having a "strong", "moderate", "limited", "inconsistent", or "insufficient" level of evidence to predict RTW in MSD and CMD populations separately. The level of evidence was attributed by counting the number of multivariate effects tested that were statistically significant ($p < .05$) with a positive (+ = protection factor) or negative (– = risk factor) association with the outcome. The detailed evidence-synthesis rules are documented in Fig 2, from Villotti and colleagues [17, 20], allowing to set the level of evidentiary support as follows: 1) "strong", when three or more studies were found statistically significant, or the ratio was between 80 and 99.9%; 2) "moderate", when two effects were found, or the ratio was between 65 and 79.9%; 3) "limited", when only one effect (positive or negative) was found, or the ratio among significant and non-significant evidences was between 60 and 64.9%; 4) "inconsistent", when the studies did not meet the criteria for any level of evidence and there was no consistent agreement in reported outcomes; 5) "insufficient", when information was not inconsistent but did not meet the criteria for limited evidence (Fig 2).

## Phase 2 –Identification and description of the measurement tools

**Inventory of tools.**  The inventory of tools was made for each prognostic factor of RTW reaching a moderate or strong level of evidence. The first article that ever validated the tool was first considered. We also considered reviews that summarized the psychometric properties of the tool. Thus, no systematic search was performed in the databases for all the studies substantiating the different psychometric properties of the tools.

**Table 2. Strategy to appraise the measurement tools.** The overall rating (Excellent, Good, Questionable) is depending on the number of psychometric (n = 6) and usability (n = 4) criteria that were assessed positively.

| | | Psychometric criteria | | |
|---|---|---|---|---|
| | Number of criteria met | 5–6 on 6 | 3–4 on 6 | 2 or less on 6 |
| Usability criteria | 4 on 4 | Excellent | Excellent | Questionable |
| | 3 on 4 | Excellent | Good | Questionable |
| | 2 or less on 4 | Good | Good | Questionable |

**Critical appraisal of the tools.** Tools were critically appraised according to six psychometric (scientific) and four usability (practical) criteria, the latter being the ones that a practitioner would also need to look at for their standard practice. The psychometric properties were as follows: 1) face validity; 2) construct validity; 3) convergent validity; 4) internal consistency; 5) test-retest reliability; 6) predictive validity. The usability criteria (practical relevance) were as follows: 1) time of assessment, 2) administrative burden, 3) ease of interpretation and 4) accessibility. These criteria were considered as sufficient based on a consensus involving 11 researchers and 12 RTW stakeholders, namely three representatives each from healthcare professionals, employers, unions and insurers [26]. The operationalization of these psychometric and usability criteria is detailed in S1 File, and were used by three authors (PV, A-C K, CL) to evaluate the tools retained from prognostic RTW factors reaching moderate to strong evidence in the review. An overall evaluation of the tool being "excellent", "good" or "questionable" was obtained by crossing the psychometric score with the practical one (Table 2). At least two attempts were made to contact some of the authors to obtain more specific information about some of the tools used (item wording, scales, scoring) but these attempts were unsuccessful in the majority of cases.

## Results

A full breakdown of article identification and selection procedure for this review is outlined in Fig 3. A total of 10981 articles were identified through the database searches. Following the removal of duplicates, 6223 references remained. A further 5364 were excluded based on title and abstract. Full texts for the remaining 877 articles were retrieved. After full text review, 78 articles remained for data extraction. The main characteristics of these 78 included studies are reported in S2 File. At full text revision stage, articles were excluded for several reasons, such as language (i.e., other than English or French), study design (e.g., qualitative study), publication type (e.g., review, dissertation thesis), population (i.e., other than MSDs or CMDs), statistics (e.g., not controlling for age and sex), outcome (i.e., RTW measures other than those stated in our inclusion criteria), predictor (e.g., not measured at baseline, not modifiable).

Table 3 provides a general indication of the amount of research on which the level of evidence for each factor is based. More particularly, it reports the number of factors measured for MSDs and CMDs, as well as the proportion of these factors that were measured in only 1, 2, 3 to 5, or more than 5 studies. A total of 90 factors were identified for MSDs, and a total of 46 for CMDs. It was conceptually acceptable to merge some concepts to produce 19 and 7 additional factors, respectively. These additional factors either combine closely related concepts (e.g., *All work accommodations factors*, combining work accommodations offer/availability/feasibility, worksite visit, workload, RTW plan, and work accommodations general; *All CMD symptoms*, namely anxiety, cognitive difficulties, depression, stress; or *All participation factors*, combining the importance to participate to family, leisure, and work activities), or were measured with different tools (e.g., *All activities disability questionnaires*, using the Oswestry Disability Index (ODI), Quebec back pain disability scale (QBPDS), Roland-Morris disability questionnaire

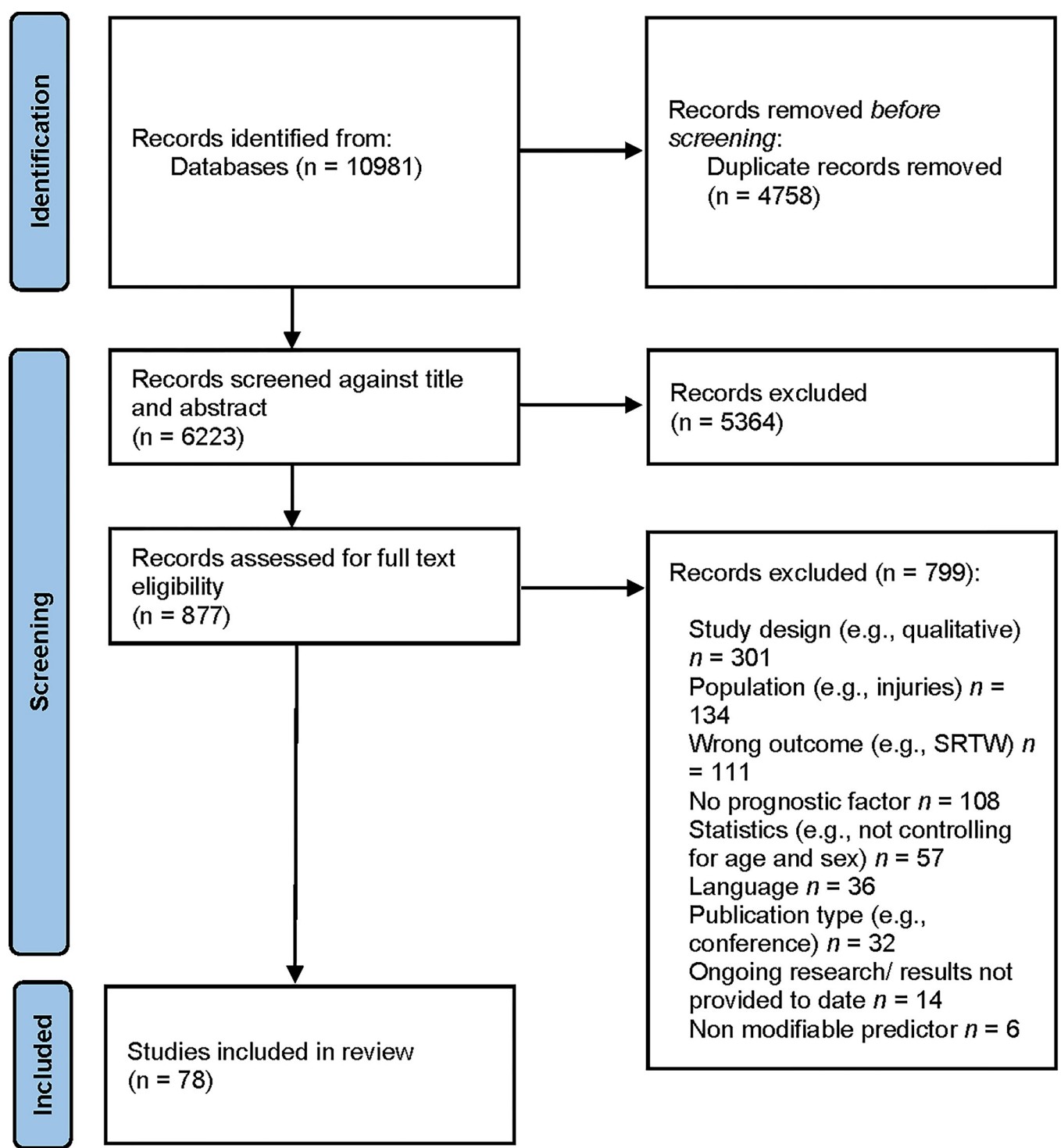

**Fig 3. Results of the search strategy (PRISMA flowchart: https://prisma-statement.org/prismastatement/flowdiagram.aspx).** The same publication can investigate organizational, personal work-related and personal non-work-related factors; therefore, the sum of the publications on these different categories of factors is higher than the number of publications that met eligibility criteria.

**Table 3. Number of factors measured for MSDs and CMDs.**

|  | MSDs | CMDs |
|---|---|---|
| Work-related organizational factors | 21 | 9 |
| Work-related personal factors | 25 | 15 |
| Non-work-related personal factors | 25 | 15 |
| Additional merged factors | 19 | 7 |
| Total number of factors | **90** | **46** |
| Factors measured in one study | 53/90 (59%) | 30/46 (65%) |
| Factors measured in two studies | 21/90 (23%) | 10/46 (22%) |
| Factors measured in 3 to 5 studies | 24/90 (27%) | 15/46 (33%) |
| Factors measured in > five studies | 26/90 (29%) | 3/46 (7%) |

(RMDQ), and other disability questionnaires). For both MSDs and CMDs, the majority of factors were measured in just one study each (i.e., 59% and 65% respectively), while a smaller proportion (29% and 7%) were measured in more than five studies (Table 3).

Table 4 reports the prognostic factors that have reached strong, moderate, and limited levels of evidence for MSDs and CMDs (more detailed in the S3 File). For MSDs, seven factors reached strong evidence, namely *Work accommodations (offer/availability/feasibility)*, *All work accommodations factors* which is a merged factor, *Expectation (RTW)*, *Fear (Fear Avoidance Questionnaire, work subscale)*, *All coping strategies factors*, *Expectations (recovery)* and *Locus of control*. Thirteen factors reached moderate evidence, namely *Job demands (physical)*, *Job strain*, *Work ability*, and *Self-efficacy (RTW)*, *Referred pain (back pain)*, *Activities (disability/ODI)*, *Activities (disability/SF-36)*, *Catastrophizing (pain)*, *All fear factors*, *Illness behaviour*, *Mental vitality*, *Positive health change* and *Sleep quality*. For CMDs, *Expectation (RTW)* emerged as the only factor reaching strong evidence, while *Job strain*, *Job demands (psychological)*, *Sleep quality* and *All participation factors* reached moderate evidence. Factors that reached a limited, inconsistent, or insufficient level of evidence for MSDs and CMDs are reported in S4 File.

Measurement tools for the prognostic factors reaching strong, moderate, and limited levels of evidence for MSDs and CMDs ranged from single-item tools to multi-item standardized questionnaires or subscales. Table 5 summarize this information. The detailed description (i.e., the title and number of items, scales and scoring, accessibility) of the tools used to measure the factors that reached strong and moderate evidence is available in S5 File. S5 File provides the information about the psychometric properties of the tools (6 criteria), their usability (4 criteria) and their global appraisal (i.e., excellent, good, questionable). They are ordered according to their level of evidence (strong evidence first) and then in a sequence allowing to make relationships between some concepts, as presented in the discussion section. For a given factor, the measurement tools ranged from single-item to multi-item tools (questionnaires) or interviews to direct measures during the clinical physical examination.

## Discussion

The main results of this study cover three areas: (1) more prognostic factors reached moderate or strong evidence for MSDs (n = 19) than CMDs (n = 5); (2) each of these factors were measured with tools (between one and 14) having different psychometric properties and usability; and (3) limited or insufficient evidence was obtained for a large proportion of prognostic factors that were seldom studied.

As discussed in the "Strengths and limitations" section, considering specific methodological aspects (e.g., inclusion criteria) and the assessment of study quality (RoB), only the factors that

**Table 4. Summary of support for the MSDs and CMDs modifiable factors reaching strong, moderate or limited level of evidence[a].**

| Strong evidence | Moderate evidence | Limited evidence |
|---|---|---|
| **Body functions and structures** | | |
| | *Referred pain (back pain)*: MSDs = 3-/4; Neg.: [27–29]; Pos.: none; N.S.: [30] | *Body functions (aerobic capacity)*: MSDs = 1-/1 [31] |
| | *Sleep quality*: MSDs = 2+/3; Neg.: none; Pos.: [32, 33]; N.S.: [34] CMDs = 2+/2 Neg.: none; Pos.: [32, 35]; N.S.: none | *CMD symptoms (depression/HADS)*: MSDs = 1-/1 [36] |
| | | *Fatigue*: CMDs = 1-/1 [35] |
| | | *Pain during activities*: MSDs = 1-/1 [37] |
| | | *Pain recurrence*: MSDs = 1+/1 [38] |
| **Activities & Participation** | | |
| | *Activities (disability/ODI)*: MSDs = 2-/2 Neg.: [39, 40]; Pos.: none; N.S.: none | *Activities (FCE)*: MSDs = 1-/1 [41] |
| | *Activities (disability/SF-36)*: MSDs = 2-/3 Neg.: [31, 43]; Pos.: none; N.S.: [44] | *Importance participation (family)*: MSDs = 1-/1 [42] |
| | *All participation factors*: CMDs = 2+/3 Neg.: none; Pos.: [35, 45]; N.S.: [43] | *Importance participation (leisure activities)*: MSDs = 1-/1 [42] |
| | | *Importance of participation (at work)*: MSDs = 1+/1 Neg.: none; Pos.: [42]; N.S.: none |
| | | *Participation (general)*: CMDs = 1+/1 [45] |
| **Work-related environmental (organizational) factors: Task contents** | | |
| | *Job demands (physical)*: MSDs = 11-/16 Neg.: [32, 34, 46–49]; Pos.: none; N.S.: [50–54] | |
| | *Job demands (psychological)*: CMDs = 2-/3 Neg.:[55, 56]; Pos.: none; N.S.: [57] | |
| | *Job strain*: MSDs = 3-/4; Neg.:[55, 56, 58]; Pos.: none; N.S.: [59] CMDs = 2-/2; Neg.: [55, 56]; Pos.: none; N.S.: none | |
| **Work-related environmental (organizational) factors: Terms of employment** | | |
| *Work accommodations (offer/availability/feasibility)*: MSDs = 4+/4; Neg.: none; Pos.: [60–63]; N.S.: none | | *Work accommodations (worksite visit)*: MSDs = 1+/1 Neg.: none; Pos.: [61]; N.S.: none |
| *All Work accommodations factors*: MSDs = 6+/7 Neg.: none; Pos.: [47, 60–63]; N.S.: [64] | | *Work accommodations (RTW plan)*: MSDs = 1+1 Neg: none; Pos.:[47]; N.S.: none |
| | | *Terms employment (contract)*: CMDs = 1-/1 Neg.: [51]; Pos: none; N.S.: none |
| **Work-related environmental (organizational) factors: Social relationships at work** | | |
| | | *Communication stakeholders (worker/supervisor)*: CMDs = 1+/1; Neg.: none; Pos.: [65]; N.S.: none |
| | | *Social support (mix of work and outside)*: CMDs = 1+/1 Neg.: none; Pos.: [35]; N.S.: none |
| | | *Social support (supervisor)*: MSDs = 1+/1 Neg.: none; Pos.: [66]; N.S.: none |
| **Work-related personal factors** | | |
| *Expectation (RTW)*: MSDs = 14+/15; Neg.: none; Pos.: [27, 32–34, 59, 63, 67–74]; N.S.: [62] CMDs = 6+/7; Neg.: none; Pos.: [32, 69, 75–78]; N.S.: [72] | *Self-efficacy (RTW)*: MSDs = 2+/2; Neg.: none; Pos.: [51, 66]; N.S.: none | *Attorney Involvement*: MSDs = 1+/1 Neg.: none; Pos.: [79]; N.S.: none |
| | | *Fear (Fear Avoidance Belief Questionnaire, work subscale)*: CMDs = 1-/1 Neg.: [80]; Pos.: none; N.S.: none |
| *Fear (Fear Avoidance Belief Questionnaire, work subscale)*: MSDs = 7-/8 Neg.:[30, 40, 52, 58, 63, 71, 80]; Pos.: none; N.S.: [81] | *Work ability*: MSDs = 3+/4 Neg.: none; Pos.: [69, 72, 82]; N.S.: [37] | *Injustice (organizational)*: MSDs = 1-/1 Neg.: [66]; Pos.: none; N.S.: none |

*(Continued)*

**Table 4.** (Continued)

| Strong evidence | Moderate evidence | Limited evidence |
|---|---|---|
| | | *Job attitudes (work importance)*: MSDs = 1+/1<br>Neg.: none; Pos.: [83]; N.S.: none |
| | | *Language barriers*: MSDs = 1-/1<br>Neg.: [79]; Pos.: none; N.S.: none |
| | | *Self-efficacy (functional)*:<br>MSDs = 1+/1; Neg.: none; Pos.:[32]; N.S.: none<br>CMDs = 1+/1; Neg.: none; Pos.: [32]; N.S.: none |
| | | *Work ability expectations*: CMDs = 1+/1<br>Neg.: none; Pos.: [84]; N.S.: none |
| Personal factors: General 'mental' personal factors / psychological assets | | |
| *Expectations (recovery)*: MSDs = 6+/6<br>Neg.: none; Pos.: [33, 37, 44, 85–87]; N.S.: none | *Catastrophizing (pain)*: MSDs = 3-/4<br>Neg.: [29, 48, 88]; Pos.: none; N.S.: [71] | *Coping (cognitive strategies)*: MSD = 1-/1 [89] |
| | | *Coping (behavioural strategies)*: MSD = 1-/1 [90] |
| *All coping strategies factors*: MSD = 3-/3<br>Neg.: [89–91]; Pos.: none; N.S.: none | *All fear factors*: MSDs = 10-/15<br>Neg.: [29–31, 37, 40, 58, 66, 67, 71, 80]; Pos.: none; N.S.: [30, 33, 80, 81, 88] | *Fear (FABQ-total score)*: MSDs = 1-/1 [58] |
| *Locus of control*: MSDs = 3+/3<br>Neg.: none; Pos.: [37, 82, 92]; N.S.: none | *Illness behaviour*:<br>MSD = 3-/4; Neg.: [28, 44, 93]; Pos.: none; N.S.: [59] | *Fear (work/health interference)*: MSDs = 1-/1 [66] |
| | *Mental vitality*: MSDs = 3+/4<br>Neg.: none; Pos.: [28, 44, 92]; N.S.: [43] | *Psychological flexibility*:<br>MSDs = 1+/1 [32]; CMDs = 1+/1 [32] |
| Personal factors: Disease related factors (including comorbidity) | | |
| | *Positive health change*:<br>MSD = 3+/4; Neg.: none; Pos.: [28, 43, 87]; N.S.: [44] | *Drug dependence*: MSDs = 1-/1 [94] |
| | | *General Health*: CMDs = 3+/5<br>Neg.: none; Pos.: [43, 75, 95]; N.S.: [69, 96] |
| | | *Seeking treatment*: MSDs = 1-/1 [70] |
| | | *Somatic*: MSDs = 1-/1 [97] |
| Personal factors: Lifestyle (habits) | | |
| | | *Lifestyle (physical activity)*: MSDs = 1+ [82] |

[a] (+): statistically significant positive association; (-) statistically significant negative association; (NS) non statistically significant association. Rules to determine the level of evidence are reported in Fig 2. Example of interpretation: "*Job demands (physical)* 11–/16", seen in the "Moderate evidence" column, means that physical job demands reached moderate evidence of a negative association with RTW (risk factor), as 11 out of the 16 studies assessing this factor showed this negative association.

have reached moderate or strong evidence will be discussed hereafter. It was also elected not to compare our results to other reviews as large methodological differences (e.g., workers with MSDs or CMDs; inclusion of prognostic studies only; specific RTW outcomes; adjustment for age and sex in multivariate analyses) make these comparisons irrelevant.

## Overall trends about prognostic factors of RTW and corresponding measurement tools

In the current state of research, more attention was paid to personal factors rather than organisational ones, which could lead to an ascription of the responsibility to the individual rather than the workplace [98]. However, as concluded in a systematic review on RTW interventions for MSDs and CMDs, successful RTW solutions are made of intervention components from three domains, namely healthcare provision (worker focused), service coordination (e.g., return to work coordinator) and work accommodations (workplace-focused) [99].

**Table 5. Summary of prognostic factors' measurement tools.**

| Factor | Single item | Two or more items | Items from a standardized questionnaire | Subscale of a standardized questionnaire | Multi-item standardized questionnaire | Other (e.g., physical evaluation) |
|---|---|---|---|---|---|---|
| Work accommodations | 3 | 1 | 1 | 0 | 0 | 0 |
| Expectations (RTW) | 9 | 0 | 3 | 1 | 1 | 0 |
| Fear (FABQ-W) | 0 | 0 | 1 | 2 | 0 | 0 |
| All coping strategies factors | 0 | 0 | 0 | 3 | 0 | 0 |
| Expectations (recovery) | 3 | 1 | 1 | 0 | 1 | 0 |
| Locus of control | 0 | 0 | 0 | 3 | 0 | 0 |
| Job demands (physical) | 0 | 1 | 2 | 2 | 0 | 1 |
| Job strain | 0 | 0 | 0 | 1 | 0 | 0 |
| Work ability | 0 | 0 | 2 | 0 | 1 | 0 |
| Self-efficacy (RTW) | 0 | 0 | 0 | 0 | 2 | 0 |
| Referred pain (back pain) | 1 | 0 | 0 | 0 | 0 | 2 |
| Activities (disability/ODI) | 0 | 0 | 0 | 0 | 1 | 0 |
| Activities (disability/SF-36) | 0 | 0 | 0 | 2 | 0 | 0 |
| Catastrophizing (pain) | 0 | 0 | 0 | 1 | 1 | 0 |
| All fear factors | 2 | 0 | 0 | 2 | 0 | 0 |
| Illness behaviours | 0 | 0 | 0 | 0 | 0 | 3 |
| Mental vitality | 0 | 0 | 0 | 1 | 0 | 0 |
| Positive health change | 0 | 0 | 1 | 0 | 0 | 0 |
| Sleep quality | 2 | 0 | 0 | 0 | 1 | 0 |
| Psychological job demands | 0 | 0 | 0 | 1 | 0 | 0 |
| All participation factors | 0 | 0 | 1 | 0 | 1 | 0 |
| **Total number of tools** | **20** | **3** | **12** | **19** | **9** | **6** |

Note: Please refer to S5 File for the identification and a detailed appraisal of the tools

There were more prognostic factors reaching moderate or strong evidence for MSDs than CMDs, which can be attributed to the much lower number of prognostic studies on CMDs. A review of reviews observed that most of the literature about factors influencing the RTW in relation to specific diseases (MSDs, CMDs, cardiovascular diseases and cancers) concerned MSDs [18]. Even if nowadays, CMDs are one of the leading causes of disability, it has been difficult to have them recognised and compensated as an occupational disease. Consequently, research on RTW after a CMD was initiated latter and call for more high-quality studies.

Except for *Referred pain (back pain)* and *Sleep quality*, none of the numerous candidate factors (e.g., MSD or CMD symptoms, reflexes, muscle strength/endurance, joint mobility/flexibility) of the ICF category of body functions and structures reached moderate or strong evidence, which concurs with analyses (structural equation modeling) showing no direct link between body functions/structures and participation (here work participation) [100].

Many tools described in the present review were single items taken from a standardized questionnaire or were simply self-developed by researchers. Single-item tools might be adequate to measure simple constructs but were automatically (and possibly erroneously) downgraded here in terms of psychometric properties as the assessment of construct validity required a factor analysis applied on several items and internal consistency can be assessed only on two item-tools and over. Consequently, the reader is advised not to judge the psychometric qualities of these tools too severely, as they can still be sensitive enough, especially if the scales comprise several points of discrimination and their inter-rater and/or intra-rater (test-retest) reliability is demonstrated (https://measuringu.com/single-multi-items/). Although these tools do not appear psychometrically sound at first sight, they are admittedly attractive to clinicians in terms of usability as this is clearly a matter of time and supporting resources [101]. Physicians and occupational therapists prefer discussing factors with the help of a topic list instead of using standardized tools [102]. These single items could thus be part of this topic list as they have at least shown their predictive validity, which is better than any other single item that have not been subjected to any form of validation. The information available in the S5 File allows for an informed choice to balance psychometric and usability criteria, when more than one tool is available.

## Organizational factors

***Work accommodations (offer/availability/feasibility)*** **and** ***All work accommodations factors*** **(for MSDs).** In total, only one organizational factor reached strong evidence, and only for workers on sick leave due to MSDs. The offer or availability of work accommodations (e.g., light duties, less working hours [63]) was evaluated as facilitating RTW, which applied specifically for employees with MSDs. Results are in line with the current state of research underlining work accommodations as an important pillar of disability management at work depending on company specific policies and programmes [103, 104] and represent one of the three main intervention domains (healthcare provision; service coordination; work accommodations) that makes RTW interventions successful for both MSDs and CMDs [99]. When summarizing all types of work accommodations (by adding for instance the workload or worksite visits), a strong level of evidence was achieved for employees with MSDs, which further support these reviews.

Nevertheless, when interpreting the results on work accommodations, it should be taken into account that included studies mainly relied on single items with poor psychometric characteristics. Therefore, upcoming research should strengthen the development of new tools or adapting the existing ones, like the Work Accommodation and Natural Support Scale (WANSS; [105]) or the Job Demands and Accommodation Planning Tool (JDAPT; [106]).

***Job demands (physical), job demands (psychological)*,** **and** ***job strain.*** In terms of other organisational factors like task contents, physical job demands for employees with MSDs and psychological job demands for employees with CMDs reached moderate evidence. This was further substantiated in a recent prospective study of 55467 employees as an exposure to a combination of different types of job demands (e.g., high quantitative, unclear and contradictory demands) that was linked to an increased risk of long-term sickness absence, manifested through additive or super additive effects [107]. Given certain preconditions, psychological job demands may also result in increased motivation at work, personal learning, or development [108]. However, during the RTW process, high perceived physical and/or psychological job demands may lead to adverse health complaints, cause fears of relapse or worsening of symptoms, affecting the employees way back to work indirectly [56, 109]. The present review also underlined that high job strain, or mentally strenuous work, which combines high

psychological (or psychosocial) job demands and low job control (Job-Demand-Control-Model, [110]), showed moderate evidence for both employees with MSDs and employees with CMDs. As discussed elsewhere [111], work stress can either boost behaviours as smoking or lack of exercise or can involve various mechanisms generating imbalance in various stress-sensitive systems, negatively impacting various diseases, including MSDs and CMDs.

The examination of physical job demands was conducted with both single items and subscales of validated questionnaires. Unfortunately, in two studies, it was not possible to find specific information on which items were selected from the Dutch Musculoskeletal Questionnaire to make up certain tools, which calls for more precise description to inform future work (new research or reviews). Both psychological job demands, and job strain were assessed using some subscales of the Job Content Questionnaire, which shows excellent psychometric properties but has some usability issues (e.g. accessibility). However, those approaches to assess job strain or job demands may introduce biased results due to self-reported answers, wherefore objective approaches were requested in the current state of research [48].

## Personal factors

*Expectations (RTW)* **(for MSDs and CMDs) and** *expectations (recovery)* **(for MSDs).**   In the present review, rather than merging them as a broader concept as previously reviewed [112, 113], RTW expectations, referring to work participation, were differentiated from recovery expectations, referring to health and healing [114]. RTW expectations reached a strong level of evidence for both MSDs and CMDs, indicating that employees having optimistic expectations towards RTW reached positive outcomes. Recovery expectations are also thought to impact the clinical outcome of several health conditions via behavioural and physiological mechanisms [115]. RTW expectations is the most studied prognostic factor (here 14+/15 for MSDs and 6+/7 for CMDs), also showing highly consistent predictive findings. Consequently, exchanging whether an employee is expecting to RTW irrespective of the health condition, may recognise those at risk for delayed RTW, especially at the beginning of sick leave. However, RTW expectations being a very complex construct (as discussed in [116] and in [112]), clinicians should be trained to recognise the factors influencing RTW, especially in view of ongoing stigma and discrimination of employees with CMDs and their capabilities of RTW [14].

Both concepts (RTW expectations, recovery expectations) are considered as complex constructs [112], thus calling for measurement tools comprising several dimensions and items, as recommended [117]. Yet, most studies used single items to assess RTW expectations with little consistency on the measurement (e.g., on terminology, timeframes for RTW or return to different types of duties) leading to limited comparability of results [118]. Likewise, as in the review of Ebrahim, Malachowski [112] on measures of patients' expectations about recovery, tools to measure recovery expectations were numerous in the present review, but including primarily single items (4 studies) or a non-standardized questionnaire without scoring information (2 studies), all leading to a low psychometric score of 2/6. Interestingly, single-item tools measuring RTW expectations were shown as predictive of RTW outcomes as multi-item tools [113]. Unfortunately, this was not shown for recovery expectations as there were no corresponding multi-item tools for this concept in Carriere, Pimentel [113]' review. However, another review identified the Brief Illness Perceptions Questionnaire as a good potential candidate [114].

Most of the studies did not assess the internal consistency based on a variation of items covering multiple domains of an underlying construct [112]. The only RTW expectations scale proposed in the current state of research was the Work-Related Recovery Expectations Questionnaire [68, 74, 119] with some limitations on psychometric characteristics. In conclusion,

considering the highly consistent predictive value of RTW expectations and recovery expectations [112, 113], the development of a multi-dimensional/item tool seems justified, especially to help better identifying intervention targets within these complex constructs. For example, different aspects of RTW expectations should be considered, like the modification of work, hierarchies, expectations to deal with job demands or those about pre-absence levels of productivity [118].

***All fear factors*** **and** *fear (Fear-Avoidance Beliefs Questionnaire—Work subscale)*, **both for MSDs.** ***All fear factors*** combine fears about work-related and non-work-related activities (e.g., physical activities) and it is part of the fear-avoidance model. Fear of relapse or movement can lead to a maladaptive strategy of activity avoidance, which reduces anxiety in the short term (related to activities likely to cause pain), but maintains the fear (as not exposed to these activities), which results in disability over time [120]. Moreover, fear of pain has been shown as more hindering than pain intensity itself, leading to adaptions in behaviour like limitations while moving or when fulfilling certain activities [30, 31]. Therefore, interventions aiming at decreasing fear-avoidance, involving gradual exposure to feared activities at work or a reduction of job demands were described as appropriate strategies for patients with MSDs [71, 80], as well as cognitive behavior therapies and psychoeducation strategies designed to target patient-specific fears [121].

The studies supporting the ***All fear factors*** (10 out of 15) either used one item (Tampa Scale for Kinesiophobia, Graded Reduced Work Ability Scale) or one subscale (4-item Fear of relapse subscale of the Return-to-Work Obstacles and Self-Efficacy Scale; 7-item Work subscale of the Fear-Avoidance Beliefs Questionnaire or FABQ-W) of standardized questionnaires. Interestingly, seven out of the 10 studies showing a significant negative relationship with RTW used the FABQ-W, enabling the conduct of this tool-specific review, which demonstrated strong evidence for employees with MSDs. The Fear-Avoidance Beliefs Questionnaire (FABQ) was rated as excellent, with good psychometric and practical characteristics. Interestingly, among four questionnaires measuring fear-avoidance constructs, it was shown that the FABQ and the Pain Catastrophizing Scale (further discussed below) provide complementary and relevant information [122].

***Work ability*** **(for MSDs).** Work ability was broadly defined as the workers capacity (including physical, psychological, and social capabilities) to perform their work while also meeting their job demands [123]. Initially developed in the 1990s in Scandinavia, no common definition was available in the current state of research resulting in varying theoretical concepts [124]. As a result, different viewpoints on work ability are influencing available results and unclear defined concepts cause heterogeneity in measurement approaches impeding interpretation and comparison [124]. Work ability was predominantly measured by two questionnaires, namely the Work Ability Index (WAI) and the Graded Reduced Work Ability Scale, either by the full scale or single items. While the WAI considers the employees job demands, current health status and available resources, the Graded Reduced Work Ability Scale addresses other dimensions including the ability to carry out ordinary or other work in light of the complaints, the amount in which activities and well-being are affected as well as the effect on complaints when continuing to work. Additionally, other broader symptoms affecting well-being and health are assessed [82].

Overall, the WAI and the Graded Reduced Work Ability Scale were evaluated as having excellent and good psychometric properties, respectively. However, the WAI was criticized for its broad variety of questions like the number of diagnoses adding weight for those not being related to work ability [125]. Due to theoretical and practical reasons, the WAI was often replaced by a single item, which was validated by comparing it to the full version of the WAI [125].

***Self-efficacy (RTW)* for MSDs.** In the context of RTW, self-efficacy refers to the perception of obstacles at work and beliefs in own abilities to overcome them. Consequently, employees reporting higher levels of RTW self-efficacy are more likely to deal with job demands and to fulfil their role at work [66, 126]. The moderate level of evidence for workers with MSDs on RTW self-efficacy was also replicated in a review on sustained RTW when staying for a period of at least 3 months [16]. Hence, assessing RTW self-efficacy as a measure of expectancies about the work and health situation could be used as a proxy for RTW during therapy [127] or in RTW interventions as Lagerveld and colleagues [128] underlined a predictive value of self-efficacy change in terms of RTW after receiving cognitive behavioral therapy (CBT). Though, focussing on achievable objectives, workplace accommodations and gradual RTW might be necessary in the first place for patients with reasons for low levels of RTW self-efficacy, e.g., when goals were set outside their abilities [128].

The measurement of RTW self-efficacy was conducted by using either the Return-to-work Self-Efficacy Questionnaire (11 items, [126]) or the Return-to-Work Obstacles and Self-Efficacy Scale (ROSES) (46 items, [66]). Both questionnaires scored high on psychometric criteria but differed in length as the ROSES includes ten conceptual dimensions with possible RTW barriers following an assessment of capabilities in dealing with them [66].

***All coping strategies factors* (for MSDs) and *Catastrophizing (pain)* (for MSDs).** Coping refers to "cognitive and behavioural efforts to master, reduce, or tolerate the internal and/or external demands that are created by the stressful transaction" [129]. However, coping was a difficult factor to assess/define as there are apparently over 100 coping taxonomies and 400 ways of coping [130]. Following the main tools that were used in the present prognostic studies (e.g., Utrecht Coping List—UCL, Coping Strategies Questionnaire—CSQ, Chronic Pain Coping Inventory—CPCI), coping was separated in three factors: 1. *Coping (style)* using the UCL, 2. *Coping (cognitive strategies)* and 3. *Coping (behavioural strategies)*, the latter two being measured in the different subscales of the CSQ and CPCI. Strong evidence was reached when combining the evidence from the latter two factors, considering all coping strategies (cognitive and behavioural). Unfortunately, although the CSQ and CPCI questionnaires show good psychometric properties, they are not freely accessible, which represents a non-negligible barrier. *Catastrophizing (pain)* is comprised in the CSQ as this can be considered as a maladaptive cognitive coping strategy that may elicit assistance or empathic responses from others. It is currently defined as "an exaggerated negative mental set brought to bear during actual or anticipated painful experience" [131]. However, we elected to keep this factor apart as there is a debate about whether catastrophizing should be viewed as a communal coping strategy, a cognitive schemata or a personality trait [132]. Nevertheless, pain catastrophizing is considered as one of the most important psychological correlates of pain chronicity and disability and is also associated with neurophysiological processes [132]. According to the fear-avoidance model [120], pain catastrophizing leads to pain-related fear (previously discussed), which in turn leads to avoidance behaviours (or *Illness behaviour*) and then to disability. It was measured with the 13-item Pain Catastrophizing Scale (rated as excellent) or the mean of three items from this standardized questionnaire (rated as questionable).

***Locus of control* (for MSDs).** Rotter [133] defines locus of control as the degree to which a person perceives an outcome as being dependent on their own actions or those of external forces, existing along a continuum from a more internalized orientation to a more externalized orientation. Bandura [134] more recently explains that locus of control should be distinguished from other constructs such as perceived self-efficacy (e.g., RTW self-efficacy), self-esteem, and outcome expectancies (e.g., Recovery expectations or RTW expectations): "*Perceived efficacy is a judgment of capability; self-esteem is a judgment of selfworth. They are entirely different phenomena. Locus of control is concerned, not with perceived capability, but*

*with belief about outcome contingencies—whether outcomes are determined by one's actions or by forces outside one's control. High locus of control does not necessarily signify a sense of enablement and well-being.*" Three tools, all classified as excellent (Table 4), were shown as predictive of RTW. Two subscales of the Multidimensional Health Locus of Control questionnaire can be used independently, namely the 6-item "change externality" [37] or the 6-item "internality" [82] subscale. The third tool is the 3-item "internal locus of control" subscale of a modified version of another standardized questionnaire, namely the Wallston's Health Locus of Control scale [92]. This modified version is justified by the fact that as for the perceived self-efficacy concept (e.g., RTW self-efficacy), locus of control should also be measured with regard to a specific context to make it a better predictor of a given behaviour, namely RTW here [92].

*Activities (disability/SF-36)* **and** *Activities (disability/ODI)* **(for MSDs).**   Although activities and participation concepts are not well differentiated in the ICF framework, the activities domain was operationalized as simple tasks/actions (e.g., mobility and daily activities of domestic life and self-care domains) while participation was defined as the ability to perform more complex social roles (e.g., leisure/social/work life situations) within a sociocultural and physical environment [135]. Consequently, even if disability can be interpreted as activities or participation [135], "disability" questionnaires are generally more in line with activities than participation. Activities were predictive of RTW only when specific tools were used such as the generic (all health conditions) physical functioning scale of the RAND-36 or SF-36 questionnaires or the more specific (low back pain) Oswestry disability index (ODI), both having an excellent overall rating. Interestingly, the Roland-Morris disability questionnaire (Activities (RMDQ), 2-/5 studies), which is as well recognized as the ODI [136], have shown insufficient evidence for predicting RTW. The items of SF-36 physical functioning scale and RMDQ are all activities while the 10-item ODI contains two participation items (8. Sex life; 9. Social life), which may help predicting RTW.

*All participation factors* **(for CMDs).**   This factor could also be called social functioning, which refers to individual's interactions with their environment and the ability to fulfill their role within such environments as work, social activities, and relationships with partners and family [137]. It has been examined with a single-item tool measuring participation in general, including social and occupational activities (Social and Occupational Functioning Assessment Scale—SOFAS) and the two-item social functioning subscale (2 items of the SF-36) standardized questionnaire measuring participation to social life activities with family, friends, neighbors, or groups. The SOFAS, including occupational activities, is expected to be predictive of RTW outcomes but the fact that the SF-36 functioning subscale was also predictive indicates that once a worker is able to participate to social activities, it is more likely to be able to reintegrates work activities.

Self-rated health was predictive of RTW as detected by two prognostic factors, more specifically *Positive health change* **(for MSDs)** and *Mental vitality* **(for MSDs)**

A *positive health change*, compared to one year ago, can be measured at the baseline assessment, and this "relative" health status was shown as a more powerful and consistent predictor of RTW than perceived pre-injury health [87]. Interestingly, in the three studies showing an association with RTW outcomes, two out of three showed a positive correlation (protective factor) when a positive health change was observed [43, 87], while one study showed a negative correlation (risk factor) when a negative health change was observed [28]. *Mental vitality* is also an indicator of good health, as it is composed of three dimensions (energy, motivation, and resilience) and has been associated with economic (including work), societal and social participation outcomes [138]. Interestingly, these two factors can be measured with the same questionnaire (RAND-36 or SF-36), using either a single item (for *Positive health change*) or multi-item subscale (*Vitality*: 4 items), but only the multi-item subscale showed an excellent overall rating, the single-item tool being questionable psychometrically.

*Sleep quality* **(for MSDs and CMDs).** Sleep quality refers to ease in falling asleep at bedtime and staying asleep during the night and is a significant component of physical and mental health, as well as overall well-being. Interestingly, working conditions (schedules and workload) can act as precipitating and perpetuating sleep disturbances [139], which in turn can interfere with activities of daily living [140] and work occupational functioning [141]. Single-item tools (overall rating: questionable) were used in two studies, one for MSDs [34] and the other for CMDs [35], measuring overall sleep quality. A standardized 7-item questionnaire (Insomnia Severity Index) was used in one study [32] comprising both populations (MSDs and CMDs), with an excellent overall rating, also measuring the impact of sleep disturbances on daily functioning and quality of life.

Two prognostic factors of RTW for MSDs were specific to workers with back pain or sciatica, namely **Referred pain (back pain)** and **Illness behaviour**. They were also the sole factors that were not assessed using questionnaires. **Referred pain (back pain)** below the knee is well-known as a poor prognostic factor of pain disability in workers with back pain [142]. The corresponding clinical tests are rapid, which also explains why it is part of the regular clinical investigations. **Illness behaviour** is defined as "*observable and potentially measurable actions and conduct which express and communicate the individual's own perception of disturbed health*" [143]. These "pain" behaviours are interpreted as out of proportion to the underlying physical disease and related more to associated psychological disturbances. This may explain why they were predictive of RTW. While there are different questionnaires to measure illness behaviours [144], only measures from the physical examination (*n* = 3) were used in the studies substantiating this prognostic factor in the present review. Waddell's signs are inappropriate (exaggerated) responses to clinical physical examination [145] while Waddell's symptoms, collected during the clinical interview, are described as not fitting general clinical experience [146]. Finally, the pain behaviour observation system [147, 148] allows to directly observe "guarding" among five pain behaviours. While the later protocol is more standardized, requiring to systematically score the different observations across various physical tests (training required), it is lengthy and not more reliable than Waddell's signs. Waddell symptoms have the advantage of being part of the history, which reduces worker assessment time, but some psychometric qualities (e.g., internal consistency, interrater and test-retest reliability) have not been tested adequately.

## Strengths and limitations

This review focused on studies with a prospective longitudinal design only, which represent the best study design to identify prognostic factors. A particular feature of this study is the rejection of studies with a high RoB, which increases the confidence in the results and thus reduces the number (or proportion) of studies that are required to reach the strong, moderate, or limited levels of evidence. On the other hand, for most of these factors, a new study will inevitably upgrade or downgrade this evidence, which emphasize the need of new high-quality prospective studies. Limited factors should be considered as valuable potential candidates in future studies, as well as factors for which only one study was available, but showing non-significant findings. This may already involve several candidate factors as up to 59% (for MSDs) or 65% (for CMDs) of the factors identified in the present review were measured in only one study. On the opposite, other factors obtained only non-significant results from more than three studies. Unless these studies were poorly conducted for any other reason, these factors might be disregarded, especially when the participant burden is of concern.

No study was rejected based on two out of the six criteria considered for the assessment of study quality (RoB), namely the loss to follow-up and the measurement of the prognostic

factor. It was deemed adequate not rejecting studies based on the quality of measurement tools as one objective of this study was to identify the variety of tools that have shown at least some (statistically significant) RTW predictive validity. This paper evaluated the tools that were shown as predictive, providing not only table on psychometric criteria, but also on practical relevance. However, a systematic search of all studies substantiating all the psychometric properties was not carried out, leading only to a first appraisal of their quality.

Other limitations also need to be acknowledged. Only factors measured at baseline were considered as a small amount of research have looked at factors measured further in the disability/recovery process. Future studies should consider measures of factors that may appear further (e.g., psychosocial factors) as disability is prolonged or measures of change (from baseline) of a given factor after an amount of time or after a given treatment, as these factors might have some additional prognostic value. The same holds true for the measurement of the RTW outcomes as all follow-ups were combined, not allowing to determine the factors that were predictive of early (e.g., 0–6 months), mid (e.g., 6–12 months) and late RTW (e.g., >12 months). Furthermore, work-related environmental and personal factors influencing RTW were assessed by using self-administered questionnaires being subject to limited objectivity. Finally, only studies published in English or French were included.

## Conclusions

In view of MSDs and CMDs as the leading causes for work disability, several environmental and personal influencing factors were identified. In the current state of research, much more prognostic factors reached moderate or strong evidence for MSDs than CMDs, reflecting the delay in CMD research.

For each factor, the corresponding measurement tools were assessed to allow an informed choice to balance psychometric and usability criteria. Among the 68 tools tested, 20 (29%) were graded as excellent, 8 (12%) as good, and 36 (53%) as questionable (4 were not graded). This review also allowed identifying factors and tools for which more research is needed, especially for CMDs. As a result, information on applied tools will be useful not only for practice when facilitating RTW but also for upcoming research, when filling gaps in the current state of research and for policymakers in developing RTW strategies.

## Supporting information

**S1 File. Tool appraisal criteria and syntax strategy.**
(DOCX)

**S2 File. Characteristics of included studies.**
(DOCX)

**S3 File. Detailed evidentiary tables for factors reaching strong, moderate or limited level of evidence.**
(DOCX)

**S4 File. Summary of evidentiary support for factors reaching inconsistent or insufficient level of evidence or showing only non-significant results.**
(DOCX)

**S5 File. Tools appraisal.**
(DOCX)

## Acknowledgments

We acknowledge the valuable contributions of Maryse Gagnon (IRSST) and Marie-Claude Laferrière (Université Laval) for their bibliometric work, and Amir Chitour, Émilie Champagne, Joëlle Cossette, Annabelle Fortin, Karolane Groulx, Benjamin Hack, Charles Plourde, Anna Ta as research assistants.

## Author Contributions

**Conceptualization:** Patrizia Villotti, Jean-Sébastien Roy, Marc Corbière, Alessia Negrini, Christian Larivière.

**Data curation:** Patrizia Villotti, Ann-Christin Kordsmeyer, Jean-Sébastien Roy, Christian Larivière.

**Formal analysis:** Patrizia Villotti, Ann-Christin Kordsmeyer, Christian Larivière.

**Funding acquisition:** Patrizia Villotti, Jean-Sébastien Roy, Marc Corbière, Alessia Negrini, Christian Larivière.

**Methodology:** Patrizia Villotti, Jean-Sébastien Roy, Marc Corbière, Christian Larivière.

**Writing – original draft:** Patrizia Villotti, Ann-Christin Kordsmeyer, Christian Larivière.

**Writing – review & editing:** Jean-Sébastien Roy, Marc Corbière, Alessia Negrini.

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
