## [Decision Letter · Decision Letter 0]

2 Apr 2024

PONE-D-23-39270Systematic review and tools appraisal of prognostic factors of return to work in workers on sick leave due to musculoskeletal and common mental disorders.PLOS ONE

Dear Dr. Villotti,

Thank you for submitting your manuscript to PLOS ONE. After careful consideration, we feel that it has merit but does not fully meet PLOS ONE’s publication criteria as it currently stands. Therefore, we invite you to submit a revised version of the manuscript that addresses the points raised during the review process.

We look forward to receiving your revised manuscript.

Kind regards,

Mohammad Ali

Academic Editor

PLOS ONE

2. We noticed you have some minor occurrence of overlapping text with the following previous publication(s) among others, which needs to be addressed:

Villotti, P., Gragnano, A., Larivière, C. et al. Tools Appraisal of Organizational Factors Associated with Return-to-Work in Workers on Sick Leave Due to Musculoskeletal and Common Mental Disorders: A Systematic Search and Review. J Occup Rehabil 31, 7–25 (2021). https://doi.org/10.1007/s10926-020-09902-1

https://www.degruyter.com/document/doi/10.1515/9783110743647-007/html

In your revision ensure you cite all your sources (including your own works), and quote or rephrase any duplicated text outside the methods section. Further consideration is dependent on these concerns being addressed.

 [This research project was Co-funded (grant #2020-0028 to PV, CL, AN, JSR, and MC)  by the Institut de recherche Robert-Sauvé en santé et en sécurité du travail (IRSST : https://www.irsst.qc.ca/en/ ) of Quebec (Canada) and the Réseau provincial de recherche en adaptation-réadaptation (REPAR : https://repar.ca/ ). These funders were not involved in any step of the research or publication process. ].  

4. We note that this manuscript is a systematic review or meta-analysis; our author guidelines therefore require that you use PRISMA guidance to help improve reporting quality of this type of study. Please upload copies of the completed PRISMA checklist as Supporting Information with a file name “PRISMA checklist”.

Reviewers' comments:

Reviewer's Responses to Questions

**Comments to the Author**

1. Is the manuscript technically sound, and do the data support the conclusions?

Reviewer #1: Partly

Reviewer #2: Yes

2. Has the statistical analysis been performed appropriately and rigorously? 

Reviewer #1: N/A

Reviewer #2: Yes

3. Have the authors made all data underlying the findings in their manuscript fully available?

Reviewer #1: Yes

Reviewer #2: Yes

4. Is the manuscript presented in an intelligible fashion and written in standard English?

Reviewer #1: Yes

Reviewer #2: Yes

5. Review Comments to the Author

Reviewer #1: Thank you for the opportunity to evaluate this manuscript. The subject matter is crucial as many

individuals of working age are affected by musculoskeletal and common mental disorders globally.

Moreover, the subject falls within the journal's scope. In general, the manuscript contains a lot of text,

making it difficult to follow the common thread, especially in the methodology and discussion

sections. The text requires a review to eliminate any extraneous information. It is also important to

exercise caution to prevent excessive use of brackets. I have outlined both major and minor comments

below.

Title: Suggestion: Tool appraisal of prognostic factors of return to work in workers on sick leave

due to musculoskeletal and common mental disorder: a systematic review

Abstract: Ok.

Keywords: OK, but why do you have numbers and bracketing before each keyword? I recommend

you to take them away.

Introduction

Overall, the introduction is well-constructed and sound. However, the excessive use of brackets

around many words should be removed. Additionally, more examples should be provided to illustrate

the types of musculoskeletal disorders (MSDs) and common mental disorders (CMDs) that are

relevant to the study. It is important to maintain consistency in the aim/objective throughout the work,

as it currently differs between the abstract and the end of the introduction.

Methods and Materials

Begin with the design. It is unclear whether this review is registered with PROSPERO and follows the

PRISMA statement. Simplify the bracketing in this section. Additionally, reference number 14 should

be included on page 12, line 218. Examples of the stakeholders involved should be provided on page

13, line 242. In Table 3, please list the references that indicate the number of factors measured for

each MSD and/or CMD.

Results

The study's objective is addressed by the results and the authors have done a good job. However, the

supplemental material includes numerous tables, which can make it challenging to follow the results.

Discussion

The length of the discussion section should be reduced to improve its comprehensibility. Additionally,

the study's total length, including limitations and conclusion sections, is currently 16 pages.

Reviewer #2: Thank you for conducting this study. It’s a very important one and has been performed and reported very diligently. However, I have a few minor comments to make that would improve the rigor of the study.

ABSTRACT

No comments

MANUSCRIPT BODY

Methods

1. Was this review registered? If not, why was it not registered?

Results

1. You stated that measurement tools ranged from single-item tools to multi-item standardized questionnaires or subscales, but you failed to summarize those tools in your results section. The readers would want to see these tools summarized as you summarized the RTW factors. I note that you have discussed these tools in the discussion section, but they were not summarized in the results.

Conclusion

1. The conclusion is too long. You need to trim it and needs to be your inference from the results.

6. PLOS authors have the option to publish the peer review history of their article (what does this mean?). If published, this will include your full peer review and any attached files.

Reviewer #1: No

Reviewer #2: **Yes: **Musa Sani Danazumi

---

## [Author Response · Author response to Decision Letter 0]

17 May 2024

In response to the Editor

We look forward to receiving your revised manuscript.

We thank the Editor for the opportunity to revise our manuscript. We hope to have addressed the major revisions requested in a way that meets yours, and the reviewers’ expectations. As requested, a marked-up copy of the manuscript that highlights changes made to the original version has been provided. 

In response to Journal requirements

1. When submitting your revision, we need you to address these additional requirements. Please ensure that your manuscript meets PLOS ONE's style requirements, including those for file naming.

Authors’ response: We made sure to address all requirements in terms of style and file naming by referring to the templates indicated. 

2. We noticed you have some minor occurrence of overlapping text with the following previous publication(s) among others, which needs to be addressed:

Villotti, P., Gragnano, A., Larivière, C. et al. Tools Appraisal of Organizational Factors Associated with Return-to-Work in Workers on Sick Leave Due to Musculoskeletal and Common Mental Disorders: A Systematic Search and Review. J Occup Rehabil 31, 7–25 (2021). 

In your revision ensure you cite all your sources (including your own works), and quote or rephrase any duplicated text outside the methods section. Further consideration is dependent on these concerns being addressed.

Authors’ response: Thank you for pointing this out. We made sure all sources, including our own, are cited in the revised version of the manuscript. We rephrased some duplicated text outside the methods section. 

For example, in the Introduction, several references have been added:

• Bellotti L, Zaniboni S, Langlois I, Villotti P. 6 Age, Mental Disorders and Work Design Factors. In: Joy B, Sophie H, Mukta K, editors. De Gruyter Handbook of Disability and Management. Berlin, Boston: De Gruyter; 2023. p. 105-24.

• Cieza A, Causey K, Kamenov K, Hanson SW, Chatterji S, Vos T. Global estimates of the need for rehabilitation based on the Global Burden of Disease study 2019: a systematic analysis for the Global Burden of Disease Study 2019. The Lancet. 2020;396(10267):2006-17. doi: https://doi.org/10.1016/S0140-6736(20)32340-0.

• Corbière M, Villotti P, Pachoud B. Chapitre 12. Maintien en emploi avec un trouble psychique. Une synthèse des écrits. Psychologie et carrières. Louvain-la-Neuve: De Boeck Supérieur; 2022. p. 221-40.

• Gragnano A, Villotti P, Lariviere C, Negrini A, Corbiere M. A Systematic Search and Review of Questionnaires Measuring Individual psychosocial Factors Predicting Return to Work After Musculoskeletal and Common Mental Disorders. J Occup Rehabil. 2021 Sep;31(3):491-511.

• Villotti P, Gragnano A, Lariviere C, Negrini A, Dionne CE, Corbiere M. Tools Appraisal of Organizational Factors Associated with Return-to-Work in Workers on Sick Leave Due to Musculoskeletal and Common Mental Disorders: A Systematic Search and Review. J Occup Rehabil. 2021 May 21;31:7-25.

As an example of rephrasing some text:

‘Individuals living with a disability or with reduced functional capacity (temporary or permanent) are potentially affected in their ability to enter, remain, or return in the workforce’ has been rephrased in ‘People with disabilities or decreased functional abilities, whether temporary or permanent, may face challenges in entering, staying in, or re-entering the workforce [2, 3]’. All changes are tracked and visible in the new submitted version of the Manuscript.

Other changes are tracked in the new version of the Manuscript.

 [This research project was Co-funded (grant #2020-0028 to PV, CL, AN, JSR, and MC) by the Institut de recherche Robert-Sauvé en santé et en sécurité du travail (IRSST : https://www.irsst.qc.ca/en/) of Quebec (Canada) and the Réseau provincial de recherche en adaptation-réadaptation (REPAR : https://repar.ca/). These funders were not involved in any step of the research or publication process. ]. 

Authors’ response: We added the phrase ‘The funders had no role in study design, data collection and analysis, decision to publish, or preparation of the manuscript’ (see page 39 of revised Manuscript) and included it in the cover letter as requested.

4. We note that this manuscript is a systematic review or meta-analysis; our author guidelines therefore require that you use PRISMA guidance to help improve reporting quality of this type of study. Please upload copies of the completed PRISMA checklist as Supporting Information with a file name “PRISMA checklist”.

Authors’ response: We submitted the PRISMA checklist as requested.

General comments to reviewers: 

We appreciate the time taken by both reviewers to evaluate this very long article. In response to their comments, we have made all possible changes to improve the readability of the article with the exception of the points related to its length. You will see that we decided to not shorten the article, and this for several reasons that you can see explained in a detailed manner in the responses below. Our choice is not made by being lazy or willing to be simply right. It took us several months of writing, leading to multiple versions sharing the right information between the paper and the supplementary files, to develop the best possible strategy. If you find better strategies, we would be happy to implement them.

In response to Reviewer #1

Reviewer #1: Thank you for the opportunity to evaluate this manuscript. The subject matter is crucial as many individuals of working age are affected by musculoskeletal and common mental disorders globally. Moreover, the subject falls within the journal's scope. 

Authors’ response: We would like to thank Reviewer #1 for the time invested in evaluating our manuscript and for all suggestions and comments provided.

In general, the manuscript contains a lot of text, making it difficult to follow the common thread, especially in the methodology and discussion sections. The text requires a review to eliminate any extraneous information. 

Authors’ response: We agree that the manuscript in its current form contains a lot of information. This situation is explained as follows. 

Initially, the manuscript’s content was divided into two separate articles:

1. Villotti, P., Kordsmeyer, A.-C., Roy, J.-S., Corbière, M., Negrini, A., Larivière, C. Systematic review and tools appraisal of prognostic factors of return to work: Part 1 – Organizational and personal work-related modifiable factors. 

2. Larivière, C., Kordsmeyer, A.-C., Roy, J.-S., Corbière, M., Negrini, A., Villotti, P. Systematic review and tools appraisal of prognostic factors of return to work: Part 2 - Personal non-work-related modifiable factors. 

However, the Editor asked us to combine the two articles into one, in order to meet the Journal’s standards. We had argued against this request, willing to keep the two articles independent, considering the huge amount of information that this review provides. Indeed, this review covers not only the prognostic factors, but also the tools for measuring them, which is infrequent, while adding so much more value to the review. We also argued that by combining the two articles, it was quite possible that the reviewers would ask to reduce its length, leading to a loose of relevant information. We were told that there was no reason for this. Consequently, we combined the two articles, eliminating as much redundancy as we could, and reducing any extraneous information. We are afraid that we cannot do more without causing damage to the overall quality of the work done. We would like to reiterate the fact that it would be regrettable to eliminate information just so that readers would not "waste" too much time reading the article… We believe that readers may be free to browse the article as they wish, focusing more on certain factors or tools and less on others, as needed. It would be unfortunate to remove useful information about some factors or tools that may be of interest to them. 

It is also important to exercise caution to prevent excessive use of brackets. 

Authors’ response: We agree. Changes have been done accordingly. 

I have outlined both major and minor comments below.

Title: Suggestion: Tool appraisal of prognostic factors of return to work in workers on sick leave due to musculoskeletal and common mental disorder: a systematic review

Authors’ response: We would prefer to keep the current tile (i.e. Systematic review and tools appraisal of prognostic factors of return to work in workers on sick leave due to musculoskeletal and common mental disorders). Because the systematic review methodology applies to the identification of the prognostic factors of RTW solely, we fear the suggested Title may be misleading (i.e. believing that the systematic review methodology applies to the tool appraisal as well, while on page 13 it reads clearly that ‘no systematic search was performed in the databases for all the studies substantiating the different psychometric properties of the tools’). 

Abstract: Ok.

Keywords: OK, but why do you have numbers and bracketing before each keyword? I recommend you to take them away.

Authors’ response: We agree. Changes have been done accordingly. 

Introduction

Overall, the introduction is well-constructed and sound. However, the excessive use of brackets around many words should be removed. 

Authors’ response: We agree. Changes have been done accordingly. 

Additionally, more examples should be provided to illustrate the types of musculoskeletal disorders (MSDs) and common mental disorders (CMDs) that are relevant to the study. 

Authors’ response: We agree. We added more examples to illustrate the types of MSDs and CMDs that are relevant to the study. On page 4, it now reads: ‘In particular, musculoskeletal disorders (MSDs, such as low back and neck pain, joint pain, tendonitis, carpal tunnel syndrome) and common mental disorders (CMDs, such as depression, anxiety, mood disorders) are leading contributors to work disability and to the global need for rehabilitation services worldwide’. We would like also to point out that detailed information about study population of retained articles is presented in Supplementary material 2, Table S1, in case the reader would need to know the full list of physical and mental disorders relevant to the study. 

It is important to maintain consistency in the aim/objective throughout the work, as it currently differs between the abstract and the end of the introduction.

Authors’ response: We agree. We made the required changes to maintain consistency throughout the manuscript. More precisely, we made modifications to the Abstract, where it now reads, ‘With the overall objective of providing implication for clinical and research practices regarding the identification and measurement of modifiable predicting factors for return to work (RTW) in people with musculoskeletal disorders (MSDs) and common mental disorders (CMDs), this study 1) systematically examined and synthetized the research evidence available in the literature on the topic, and 2) critically evaluated the tools used to measure each identified factor’.

Methods and Materials

Begin with the design. 

Authors’ response: To respond to this suggestion, at page 7, we added a paragraph entitled ‘Study design’. We now read: ‘A systematic literature review was conducted to identify prognostic factors associated with RTW among workers on sick leave due to MSDs and CMDs. The identified prognostic factors were evaluated based on their level of evidence, categorized as strong, moderate, limited, inconsistent, insufficient, or non-significant. Subsequently, the tools used to assess each identified prognostic factor, which exhibited moderate to strong evidence, were systematically described and evaluated. This evaluation encompassed an assessment of the psychometric properties of the tools, including reliability and validity, as well as their usability criteria’.

It is unclear whether this review is registered with PROSPERO and follows the PRISMA statement. 

Authors’ response: We have not registered the protocol with PROSPERO. We are aware that the practice of registering systematic review protocols has become increasingly common over the last years as part of efforts to enhance transparency, reduce bias, and improve the reproducibility of research findings. However, our detailed manuscript and supplementary files demonstrate that, despite not registering the protocol, the review has been conducted rigorously and transparently. Our work still adheres to established methodologies and standard. The extended information about methodology adopted and all supplementary files provided within our Manuscript ensures replicability. We also added the PRISMA checklist as Supporting Information in the revised submission of our Manuscript.

Simplify the bracketing in this section. 

Authors’ response: We modified the manuscript accordingly. Changes are tracked in the new version of the manuscript.

Additionally, reference number 14 should be included on page 12, line 218. 

Authors’ response: Changes have been done accordingly. 

Examples of the stakeholders involved should be provided on page 13, line 242. 

Authors’ response: Following the recommendation, we provided examples. It now reads: ‘These criteria were considered as sufficient based on a consensus involving 11 researchers and 12 RTW stakeholders (three representatives each from healthcare professionals, employers, unions and insurers).

In Table 3, please list the references that indicate the number of factors measured for each MSD and/or CMD.

Authors’ response: Table 3 is intended to provide a brief numeric overview of retained factors. We fear that, by adding all references, the readability of the Table would significantly decrease. Also, it would be a redundant information added, as the list of reference for each factor is already provided not only in Table 4, but also in each supplementary file provided alongside the Manuscript. We thus decided to not proceed with this suggestion. However, we are of course open to modify the Table if the Reviewer and the Editor still believe this represents a fundamental modification to the table to do.

Results

The study's objective is addressed by the results and the authors have done a good job. However, the supplemental material includes numerous tables, which can make it challenging to follow the results.

Authors’ response: We could not agree more with Reviewer #1 on this comment, but we sincerely don't know how to do it better. To prevent the article from being too long, we carefully selected the information that was most useful and put it in the body of the article and, we put the rest of the information in supplementary files. If Reviewer #1 have better ideas or solutions to suggest to make the article easier to read, we would be interested to hear from you.

Discussion

The length of the discussion section should be reduced to improve its comprehensibility. Additionally, the study's total length, including limitations and conclusion sections, is currently 16 pages.

Authors’ response: Again, we could not agree more with Reviewer #1 on this comment. To help readers quickly identify the information of interest to them in this lengthy discussion, we included several headings and sub-headings. To avoid unnecessarily lengthening the article, we kept the results section very short, mainly referring the reader to the results tables. Indeed, we felt it was more effective for readers to consult these tables, which are structured to quickly understand where the information is placed, rather than to describe these results in the text. On the other hand, it was necessary to quickly present this information at the be

---

## [Decision Letter · Decision Letter 1]

3 Jul 2024

Systematic review and tools appraisal of prognostic factors of return to work in workers on sick leave due to musculoskeletal and common mental disorders.

PONE-D-23-39270R1

Dear Dr. Villotti,

We’re pleased to inform you that your manuscript has been judged scientifically suitable for publication and will be formally accepted for publication once it meets all outstanding technical requirements.

Kind regards,

Mohammad Ali

Academic Editor

PLOS ONE

Additional Editor Comments (optional):

Reviewers' comments:

Reviewer's Responses to Questions

**Comments to the Author**

1. If the authors have adequately addressed your comments raised in a previous round of review and you feel that this manuscript is now acceptable for publication, you may indicate that here to bypass the “Comments to the Author” section, enter your conflict of interest statement in the “Confidential to Editor” section, and submit your "Accept" recommendation.

Reviewer #1: All comments have been addressed

Reviewer #2: All comments have been addressed

2. Is the manuscript technically sound, and do the data support the conclusions?

Reviewer #1: Yes

Reviewer #2: Yes

3. Has the statistical analysis been performed appropriately and rigorously? 

Reviewer #1: N/A

Reviewer #2: Yes

4. Have the authors made all data underlying the findings in their manuscript fully available?

Reviewer #1: Yes

Reviewer #2: Yes

5. Is the manuscript presented in an intelligible fashion and written in standard English?

Reviewer #1: Yes

Reviewer #2: Yes

6. Review Comments to the Author

Reviewer #1: Dear authors,

I would like to thank the authors for the information they provided regarding the length of the article. I agree that two articles would be more appropriate than one. However, I have nothing more to add other than that the authors have done a good job. I also recognise that important information may be lost if the article is shortened further.

Reviewer #2: Thank you for giving the opportunity to review this paper. I can now recommend this paper for publication in this journal.

7. PLOS authors have the option to publish the peer review history of their article (what does this mean?). If published, this will include your full peer review and any attached files.

Reviewer #1: No

Reviewer #2: **Yes: **Musa Sani Danazumi

---

## [Editor Report · Acceptance letter]

8 Jul 2024

PONE-D-23-39270R1 

PLOS ONE

Dear Dr. Villotti, 

I'm pleased to inform you that your manuscript has been deemed suitable for publication in PLOS ONE. Congratulations! Your manuscript is now being handed over to our production team.

Kind regards, 

on behalf of

Dr. Mohammad Ali 

Academic Editor

PLOS ONE